# Can miRNAs Be Considered as Diagnostic and Therapeutic Molecules in Ischemic Stroke Pathogenesis?—Current Status

**DOI:** 10.3390/ijms21186728

**Published:** 2020-09-14

**Authors:** Kirill V. Bulygin, Narasimha M. Beeraka, Aigul R. Saitgareeva, Vladimir N. Nikolenko, Ilgiz Gareev, Ozal Beylerli, Leila R. Akhmadeeva, Liudmila M. Mikhaleva, Luis Fernando Torres Solis, Arturo Solís Herrera, Marco F. Avila-Rodriguez, Siva G. Somasundaram, Cecil E. Kirkland, Gjumrakch Aliev

**Affiliations:** 1Department of Human Anatomy, I.M. Sechenov First Moscow State Medical University (Sechenov University), 119146 Moscow, Russia; kirill-bh-red@yandex.ru (K.V.B.); vn.nikolenko@yandex.ru (V.N.N.); 2Department of Normal and Topographic Anatomy, M.V. Lomonosov Moscow State University, 119991 Moscow, Russia; 3Department of Biochemistry, Center of Excellence in Regenerative Medicine and Molecular Biology (CEMR), JSS Academy of Higher Education and Research (JSS AHER), Mysuru 570015, Karnataka, India; bnmurthy24@gmail.com; 4Department of Neurosurgery and Medical Rehabilitation with Courses IAPE, Bashkir State Medical University, 450008 Ufa, Russia; aneuro@yandex.ru (A.R.S.); ilgiz_gareev@mail.ru (I.G.); obeylerli@mail.ru (O.B.); la@ufaneuro.org (L.R.A.); 5Research Institute of Human Morphology, 3Tsyurupy Street, 117418 Moscow, Russia; mikhalevalm@yandex.ru; 6The School of Medicine, Universidad Autónoma de Aguascalientes, 20130 Aguascalientes, Mexico; lfts99@yahoo.com.mx; 7Human Photosynthesis© Research Centre, 20000 Aguascalientes, Mexico; comagua2000@gmail.com; 8Department of Clinic Sciences, Science Health Faculty, University of Tolima, 730006299 Ibagué, Colombia; markos.avila@gmail.com; 9Department of Biological Sciences, Salem University, Salem, WV 26426, USA; siva15ram58@gmail.com (S.G.S.); EKirkland@salemu.edu (C.E.K.); 10Institute of Physiologically Active Compounds, Russian Academy of Sciences, 142432 Chernogolovka, Russia; 11GALLY International Research Institute, 7733 Louis Pasteur Drive, #330, San Antonio, TX 78229, USA

**Keywords:** microRNAs, ischemic stroke, metabolic diseases, oxidative stress, inflammation, neurogenesis

## Abstract

Ischemic stroke is one of the leading causes of death worldwide. Clinical manifestations of stroke are long-lasting and causing economic burden on the patients and society. Current therapeutic modalities to treat ischemic stroke (IS) are unsatisfactory due to the intricate pathophysiology and poor functional recovery of brain cellular compartment. MicroRNAs (miRNA) are endogenously expressed small non-coding RNA molecules, which can act as translation inhibitors and play a pivotal role in the pathophysiology associated with IS. Moreover, miRNAs may be used as potential diagnostic and therapeutic tools in clinical practice; yet, the complete role of miRNAs is enigmatic during IS. In this review, we explored the role of miRNAs in the regulation of stroke risk factors viz., arterial hypertension, metabolic disorders, and atherosclerosis. Furthermore, the role of miRNAs were reviewed during IS pathogenesis accompanied by excitotoxicity, oxidative stress, inflammation, apoptosis, angiogenesis, neurogenesis, and Alzheimer’s disease. The functional role of miRNAs is a double-edged sword effect in cerebral ischemia as they could modulate pathological mechanisms associated with risk factors of IS. miRNAs pertaining to IS pathogenesis could be potential biomarkers for stroke; they could help researchers to identify a particular stroke type and enable medical professionals to evaluate the severity of brain injury. Thus, ascertaining the role of miRNAs may be useful in deciphering their diagnostic role consequently it is plausible to envisage a suitable therapeutic modality against IS.

## 1. Introduction

Ischemic stroke (IS) is a leading cause of mortality and reporting over 6 million deaths annually worldwide [1,2]. Ischemic stroke occurs due to the cerebral blood supply disruption, which is most often due to embolism, arterial thrombosis, or arteriosclerotic changes. Worldwide, nearly 2 million cases of IS have been reported in young adults per annum [3,4,5]. Further, there are several risk factors for IS reported from epidemiological studies: The major risk factors for IS are hypertension, arterial dissections, improper lifestyle, dietary patterns, obesity, patent foramen ovale, socioeconomic status, illicit drug usage, diabetes mellitus, smoking, and hemostatic factors [3,5]. The evaluation of risk factors is an important aspect to decipher a specific subtype of IS because the prognosis, pathogenesis, and treatment are variable for different subtypes. Moreover, the selection of suitable prevention strategies to combat IS predominantly is based on the subtype of IS [6,7,8,9]. During the 21st century, the incidence of age-standardized stroke has been increasing in young adults. Older adults worldwide are affected by obesity and aging and the chances of stroke incidence among these obese individuals is reported to be increased to 1.5 million people across Europe by 2025 [9]. In addition, stroke patients are at a higher risk of acquiring cognitive abnormalities, dementia, Alzheimer’s disease, depression, and significantly mitigate the patient’s quality of life. Therefore, post-stroke therapeutic modalities to achieve immediate recovery are needed to mitigate these devastating stroke effects [3].

Arterial hypertension is the one of the significant risk factors for IS. It decreases the elasticity of blood vessels further facilitating vessel rupture, causing hemorrhagic stroke [10]. Atherosclerotic progression leads to arterial stenosis, thrombus formation, and plaque rupture, all of which can cause occlusion of arteries supplying specific brain areas, to foster ischemia [11]. Furthermore, the diabetic hyperglycemia also exacerbates neuronal death and worsens the clinical outcome in IS patients [12].

Thrombosis and embolism foster detrimental effects such as IS accompanied by the excitotoxicity, cerebral edema and blood–brain barrier (BBB) impairment. Despite the numerous therapeutic approaches and studies, thrombolysis is still the most effective treatment for IS [13,14]. Additionally, brain injury is not limited to the core ischemic zone, as dying neurons release proapoptotic and proinflammatory factors into the adjacent brain parenchyma, which lead to neuronal death in the penumbra region [14]. The ischemic penumbra region can be protected by timely treatment using therapeutic modalities that can limit infarction size, salvaging cell function and structural integrity in the penumbra region. In addition to the ischemia, reperfusion induces brain injury via multiple mechanisms including inflammation and oxidative stress [15,16]. Ischemic-type injury to developing white matter is characterized by the loss of premyelinated axons in fetal white matter and loss of axon function mediated through neurotoxic NMDA receptors, which contributes to the incidence of cerebral palsy and stroke [17]. Inflammatory processes are initiated several hours after the injury, when resident microglia and dying neurons release proinflammatory molecules. Concurrently, endothelial cells express cell adhesion molecules including intercellular adhesion molecule 1 (ICAM-1) and VCAM-1 [16]. These changes typically promote transendothelial migration of peripheral immune cells mainly macrophages and neutrophils to the ischemic region, which further potentiates inflammation by the subsequent release of pro-inflammatory molecules and reactive oxygen species (ROS). The energy deficits during brain injury induce damage to the “ion pumps across neurons”, which confer vascular-cellular edema and increase intracranial pressure [16]. Furthermore, the damaged mitochondria in brain cells release ROS, which mediates lipid peroxidation, and nucleic acids damage. All of these factors exacerbate secondary brain damage following stroke [18]. 

Inflammation and oxidative stress are significant events involved in fostering cellular and molecular damage during post-ischemic insult. However, they present two prospective therapeutic directions for the prevention of secondary brain damage after stroke [18]. Stroke initially accompanied by the acute phase followed by the chronic cellular damage conducive to the mitigation in cellular plasticity and regeneration. Based on these phases, the present therapeutic interventions are preferred in order to potentiate neuronal survival and recovery [18].

MiRNAs are the molecules reported to be involved in modulating stroke risk factors [19]. They are small non-coding RNA molecules composed of 22 nucleotides in length, reported to act as posttranscriptional regulators of gene expression in mammalian cells [12,19]. MiRNAs could modulate “base-pairing to complementary sequences” in messenger RNA (mRNA), which particularly leads to the impairment of gene activity via “translational repression”. These molecules are involved in most fundamental biological processes viz., cell cycle control, cell metabolism, apoptosis, and immune response [20]. The prevalence of IS reasonably suggests that medical professionals need to identify early signs, risk factors to choose an immediate therapeutic modality against disease processes that lead to stroke. In addition, this strategy is serendipitously beneficial for selecting a suitable therapeutic modality to treat IS pathophysiology. In this review, we have comprehensively discussed the current state of research pertaining to the role of miRNAs in cerebral ischemia and pathogenesis of IS. We focused on the pros and cons of miRNAs that modulate risk factors for “pathological mechanisms of IS” to develop diagnostic tools and novel therapeutic modalities for IS.

## 2. MicroRNAs and Risk Factors for Ischemic Stroke

A plethora of studies have outlined the role of miRNAs in regulating target genes involved in the arterial hypertension, atherosclerosis, and diabetes mellitus which predispose patients to acquire IS. The target genes are capable of mediating the inflammatory reactions, oxidative stress, cell proliferation, and apoptosis [21]. MiRNAs involved in orchestrating the IS and other risk factors were summarized in the Table 1.

Arterial hypertension is the most significant risk factor for IS, as it decreases the elasticity of arterial walls consequently lead to rupture of vessels and hemorrhagic stroke [37]. Atherosclerotic progression leads to arterial stenosis and plaque rupture, which typically forms “thrombi” to foster the obstruction of arteries supplying specific brain areas results in the progressive enhancement in ischemia [11]. Hyperglycemia during diabetes could exacerbate neuronal death in IS patients [12]. MiRNAs exhibit a significant function in modulating all of these stroke risk factors [19]. 

Past research studies described a few important miRNAs that play a key role in arterial hypertension [38,39,40,41]. For instance, MiR-155, miR-125a/b-5p, miR-22, and miR-487b can regulate blood pressure and therefore affects patient clinical outcome during IS-targeted therapy. Some miRNAs are reported to be involved in the early development of hypertension, which can be useful for early diagnosis and identifying risk predictors towards stroke incidence. A report by Yang S. et al. (2015) described the biomarkers attributed to the IS in type-2 diabetes (T2D) patients. According to this study, the hyperglycemia in T2D patients induces the platelet activation through miR-144 and miR-223 to foster the “downregulation of IRS-1 expression and upregulation in the P2Y12 expression” via IRS-1-PI3K-Akt signaling. Therefore, the low levels of platelet and plasma miR-223 expression and the increased platelet and plasma miR-144 expression may be risk factors for IS in the T2D patients [42]. Studies of diabetic-stroke using rodent models reported that the low miR-145 levels promotes “neuroprotective effect” via bone marrow stromal cell injection to improve clinical outcomes during IS [43]. 

Hyperglycemia can affect brain mRNA expression after stroke. Findings from past research studies depicted a significant link between miR-Let7A, *ASK-1* and microglial function in hyperglycemia-induced oxidative stress during stroke; these reports suggesting a need for developing pharmacological agents to enhance the microglial function against ischemic stroke [22,43,44].

### 2.1. MicroRNAs and Arterial Hypertension

MiR-155 levels exhibit a negative correlation with blood pressure in the aorta of the spontaneously hypertensive rats (SHR) [45]. It has been reported that miR-155 levels are lower in peripheral blood mononuclear cells in hypertensive patients [46]. Furthermore, miR-155 can target endothelial nitric oxide (NO) synthase and angiotensin II receptor type 1 (AT1R). This indicates its prominent role in vasorelaxation and renin–agiotensin system [47]. Thus, miR-155 is an important miRNA that can modulate IS by controlling patient’s blood pressure. 

Furthermore, several other miRNAs viz., miR-125a/b-5p, miR-22, and miR-487b could efficiently regulate blood pressure. MiR-125a/b-5p targets endothelin-1 (potent vasoconstrictor) in vascular endothelial cells whereas miR-22 targets chromogranin A involved in the “upregulation of catestatin” regulating blood pressure [38,39]. MiR-22 antagomir administration to SHR causes a reduction in blood pressure suggesting a significant therapeutic modality against IS triggered through hypertension [40]. It has been demonstrated that miR-487b is upregulated in the aorta of angiotensin II-induced hypertensive Sprague–Dawley rats; this miRNA can suppress anti-apoptotic insulin receptor 1 (IGF-I) substrate that causes damage to aortic adventitial fibroblasts [41]. MiRNA sequencing reported the differential expression of 24 miRNAs between brainstems of hypertensive SHR and normotensive Wistar–Kyoto (WKY) rats. Deep sequencing of miRNAs have identified 30 microRNAs that are upregulated in human dermal microvascular endothelial cells; these are thought to have a specific role in hypertension development followed by the incidence of IS [37,48] (Figure 1).

### 2.2. MicroRNAs and Diabetes

Diabetes-induced vascular pathologies can enhance vascular permeability to foster IS [49]. The worse poststroke outcomes were reported in mice models of diabetic stroke due to the injured white matter and axons [43,50]. Increased blood glucose level during diabetes can contribute to the development of endothelial dysfunction by affecting atherosclerotic plaque formation, which can lead to cerebral arterial stenosis, another risk factor for stroke [51]. A recent study showed that “T2D patients with IS” have significantly “downregulated expression of miR-223 and upregulated expression of miR-144” [42]. Diabetic patients with IS are also associated with a decreased platelet miR-223 and miR-146a7 expression. Low levels of these miRNAs could facilitate a wide range of stroke-induced pathophysiological changes [30]. 

Bone marrow stromal cell (BMSCs) treatment was reported to have extensive neuroprotective effects against induced transient focal ischemia in streptozotocin-induced diabetic rats compared to non-diabetic rats. This was demonstrated to be associated with the low miR-145 expression, and upregulated adenosine triphosphate-binding cassette transporter 1 (ABCA1), and insulin-like growth factor 1 (IGF-R1) in diabetic rats [43]. miR-145/ABCA1/IGFR1 pathway orchestrates the neurorestorative function of BMSCs across the injured brain region during diabetic-stroke [43]. IV injection of BMSCs derived from normal rats could induce angiogenesis, neurogenesis, and arteriogenesis followed by the vascular stabilization to enhance neuronal function after stroke in patients without diabetes [52,53,54]. BMSCs have significant functions in facilitating the release of migrating and stimulating trophic factors from brain parenchyma across the injured ischemic region of brain tissue [53,54]. 

Hyperglycemia can affect brain miRNAs expression in the absence of IS. For instance, let-7a is involved in glucose metabolism and suppresses microglial apoptosis signal-regulating kinase 1 (ASK-1) [22,44]. Another study demonstrated that the miR-200a/b and miR-466a/d-3p downregulate neural stem cells of streptozotocin-induced diabetes mice [55]. 

These studies suggest that there is a critical need to develop novel pharmacological agents to treat diabetic IS as diabetes significantly alters metabolism. Consequently, it induces changes in stroke pathology, increasing the difficulty in obtaining good clinical outcomes [43]. 

### 2.3. MicroRNAs and Atherosclerosis

Atherosclerosis is an asymptomatic inflammatory disease and a significant cause of myocardial infarction and ischemic stroke. It is initiated by the formation of atherosclerotic plaques when cholesterol with LDL accumulates in the endothelium, fostering inflammation associated with IS [56]. The atherosclerotic plaque formation is initiated by the formation of a fatty streak where lipids and immune cells are deposited in the intima [57]. Conditions such as hypertension and hypercholesterolemia are risk factors for atherosclerosis because they induce VCAM-1 expression, which enables infiltration of immune cells (monocytes and macrophages) across the ischemic region [58]. Low-density lipoproteins (LDLs) present in the fatty streak mediate the formation of necrotic core of the atherosclerotic plaques [56,59]. Atherosclerotic plaque progression is associated with multiple processes such as rapid smooth muscle cell (SMCs) proliferation, inflammation and extracellular matrix (ECM) remodeling [56].

Overexpression of miR-92a invokes the impairment of angiogenesis during atherosclerosis and enhances manifestations of ischemic stroke [60]. Blockade of miR-92a using antagomir can promote the restoration of neovascularisation across damaged tissue in vivo, and in vitro, which suggests a likely valuable therapeutic modality against ischemic disease [60]. 

Suppression of miR-320a that targets serum response factors could mediate VEGF signaling to enhance blood vessel synthesis during atherosclerosis-induced stroke [28]. The miR-143/145 can target ATP-binding cassette transporter 1 (ABCA1), whereas the miR-92a can target Kruppel-like transcription factor that modulates shear stress genes, conferring other protective effects on physiology of blood vessels during atherosclerosis [28,60,61]. MiR-145 is crucial for myocardin-mediated reprogramming of vascular smooth muscle cells (VSMC). This miRNA is capable of inducing multipotent neural crest stem cell differentiation to produce a vascular smooth muscle [61]. For instance, the miR-145 foster VSMC differentiation by enhancing Myocd protein while miR-143 and miR-145 cooperatively target Klf4 (Kruppel-like factor 4), myocardin, and Elk-1 to determine VSMC fate [61]. 

Oxidized-LDL can activate dendritic cells and enhance inflammatory responses in atherosclerosis. For instance, MiR-181a expression is suppressed in monocytes of obese patients. Treatment with miR-181a precursors can reduce LDL-associated oxidative inflammation in bone marrow-derived dendritic cells via targeting early response gene c-Fos [32,62]. Moreover, miR-155 is involved in mediating atherogenesis through inflammation by targeting anti-inflammatory transcription factor ETS1 and angiotensin II receptor type 1 (AT1r) in human endothelial cells [63]. The low MiR-155 levels in hyperlipidemic mice can facilitate the atherosclerosis progression via targeting colony stimulating factor 1 (CSF) receptors or mitogen-activated protein (MAP) kinase [64]. Macrophage-specific miR-155 has been shown to decrease the level of Fas-associated protein, and other inflammation factors such as TNF-α by targeting calcium regulated heat stable protein 1 (CARHSP1) [65,66]. Furthermore, miR-155 can target NF-κB as this transcription factor regulates miR-155 transcription in a feedback loop [67]. miR-155 is upregulated in “LDL-stimulated” human macrophages; the inhibition of miR-155 leads to the higher lipid uptake and inflammatory responses during atherosclerosis consequently risk of ischemic stroke [68]. miR-155 also promotes atherosclerosis by suppressing the activity of anti-inflammatory factors, “HMG1 protein”, “B-cell lymphoma 6 (Bcl-6) protein”, and “suppressor of cytokine signaling 1 (SOCS1)”. These studies suggest the need for developing novel pharmacological agents to impair the activity of miR-155 for blocking atherosclerosis progress, a risk factor for the development of IS [69] (Figure 1).

A cluster of miR-221/222 can target many inflammation-associated regulatory molecules in human endothelial cells, including transcription factor ETS1 which is conducive to the activation of *PGC-1α*, adiponectin receptor 1, and STAT-5. miR-221/222 cluster targets kinesin-like protein 1, 2 (Kip1, Kip2), and c-kit involved in promoting VSMC proliferation [29,70]. However, miR-221/222 levels are mitigated in advanced plaques; the complete loss of miR-221/222 could induce plaque rupture. Investigation of vascular calcification in mice showed that miR-221/222 upregulation can exacerbate aortic calcification by inducing altered expression of ectonucleotide phosphodiesterase 1 (Enpp1) and Pit-1 [71]. These reports have suggested the intricate role of miRNAs to foster the development of atherosclerosis, a well-known risk factor for IS. They envisage novel therapeutic modalities to modulate the role of above specific miRNAs in atherosclerosis as they may deliver effective clinical outcomes against IS. 

### 2.4. MicroRNAs in Brain Injury

Low blood supply during the ischemic event can cause disruption of ion gradient across the neurons, excitotoxicity, and neuronal death [72]. The intricate mechanism pertaining to brain ischemia is proteolysis and simultaneously provokes the loss of cell survival signal transduction [72]. Furthermore, ischemia leads to white matter damage with rapid release of glutamate across the neurons to mediate a heavy calcium influx through calcium channels into the neurons followed by the sustained activation of apoptotic pathways through mu-calpain, calcineurin, and lipases [17,72]. During the reperfusion period, “reoxygenation results in oxidative stress” whereas the “restoration of circulation results in progressive inflammation and edema” [72]. These events can exacerbate the vulnerability of the affected brain tissue to neurodegeneration through apoptotic mechanisms [72]. The expression levels of numerous miRNAs have been found to be altered as an aftermath of transient focal ischemic reperfusion [73] and ischemic brain injury. For instance, the transient focal ischemia in male rats is conducive to the overexpression of miR-19b, miR-290, and miR-292-5p during reperfusion [74]. However, the expression profiles of miR-150, miR-195, and miR-320 are reported to be declined in these male rats after reperfusion [74,75]. Other miRNAs viz., rno-miR-206, miR-214, miR-223, miR-298, miR-327, and miR-494 were reported to be overexpressed during ischemia/reperfusion periods in rat models [74]. The expression patterns of these miRNAs could be ascertained for possible use of miRNAs as diagnostic and prognostic biomarkers in stroke and related neuropathologies [75]. 

Furthermore, the ischemic stroke is disease subgroups viz., hyperacute ischemic stroke (HIS) group, acute ischemic stroke (AIS) group, subacute ischemic stroke (SIS) group, and restorative ischemic stroke (RIS) group [76]. Exosomal miR-21-5p, miR-30a-5p are reported to be expressed in IS models [77,78]. Exosomal miR-422a and miR-125b-2-3p expression levels are significantly declined in ischemic stroke models [76]. The exosomal miRNAs could be a promising biomarker to diagnose ischemic stroke and have clinical value to improve clinical outcomes. 

### 2.5. MicroRNAs and Ischemic Excitotoxicity

During ischemia, excessive glutamate release and defects in glutamate transporter expression typically lead to the hyperactivation of glutamate receptors which fosters neuronal death [17,79]. Overexpression of miR-223 in the hippocampus has been shown to reduce the expression levels of NR2A subunit of NMDA receptors and GluR2 subunit of AMPA receptor (AMPAR) eventually prevents neuronal death after transient global ischemia [80]. The extensive rise in circulating miR-223 levels following the stroke incidence are associated with stroke severity and infarct volume in IS patients [81,82]. Similarly, IS patients are associated with extensive exosomal miR-223 which further [23] enhanced IS severity and poor clinical outcomes [23]. miR-125b expression exacerbates excitotoxicity by increasing NMDA receptor subunit expression [25]. 

Upregulation of *NR2A* facilitates NMDA-mediated neuronal death in cortical neurons subsequently promotes IS [26]. Prevention of excitotoxicity requires rapid removal of glutamate from the synaptic cleft, which is mediated by glutamate transporters such as astrocyte glutamate transporter-1 (GLT1) [83]. There is a clear correlational evidence for the increased miR-107 and decreased GLT1 levels following cerebral ischemia [84]. For instance, the upregulation of miR-107 expression fosters excitatory neurotoxicity by inducing the downregulation in GLT-1 expression after IS [84]. Hypoxia/reoxygenation (H/R) injury to the nerve cells has provoked neuronal apoptosis due to substantial rise in both glutamate and miR-107 with the impairment of GLT-1 expression [84]. Thus, novel therapeutic modalities to inhibit miR-107 perhaps could prevent the suppression of GLT1 to clear excess glutamate accumulation and neuronal apoptosis during IS [84]. 

### 2.6. MicroRNAs and Oxidative Stress

The imbalance between free radical formation and antioxidant activity causes oxidative stress. This is a critical pathological mechanism of secondary brain injury following cerebral ischemia [85]. Both ischemia and reperfusion are characterized by the production of ROS, including superoxide free radicals, peroxide, hydroxyl radical, singlet oxygen, and peroxynitrite ions [86]. Increased ROS levels can induce damage to the neurons and promote mitochondrial dysfunction with sequential activation of calpain, inflammation, and finally neuronal death. These factors could affect infarct volume following cerebral ischemia, which predispose to IS [87] (Figure 2). 

Reports indicate that the miRNAs play an important role in regulating the oxidant-antioxidant balance after IS. For instance, Serum miR-124, miR-9, and miR-219 levels are impaired in acute ischemic stroke consequently fostering neuroinflammation and brain injury. Excessive miR-424, miR-23a-3p, and miR-99a can reduce oxidative stress consequently protecting the brain after IS [31,88]. In mice models, the administration of antagomir miR-424 following stroke typically “decreased infarct volume and enhanced the expression of *NRF-2*”, which is known to have anti-inflammatory, antioxidant, and neuroprotective effects [31]. Antagomir miR-424 treatment also enhanced the expression of mitochondrial superoxide dismutase and decreased the ROS levels. The protective effects of miR-424 against oxidative stress in neurons *in vitro* are due to the “knockdown of *NRF-2*” which confirms the antioxidant mechanism of action of miR-424 [31].

Vagus nerve stimulation (VNS) is capable of inducing neuroprotective effects during IS; this was delineated with in vivo models of ischemia/reperfusion (I/R) injury. The VNS typically could foster both neuronal and astrocyte a7nAchR activation, simultaneously blocking redox stress and neuronal death during IS. This is achieved through the “extensive expression of miR-210” and “Akt-phosphorylation” [89,90]. Thus, VNS could be a potential therapeutic modality to treat IS. However, the specific mechanism by which miR-210 facilitates protection against oxidative stress has not yet been elucidated fully. Moreover, miR-210 targets multiple mRNAs that encode proteins involved in mitochondrial function, metabolism, cell survival, and produce a pleiotropic neuroprotective effect after IS [91].

### 2.7. MicroRNAs and Alzheimer’s Disease

There is a significant correlation between cerebrovascular disease and development of Alzheimer’s disease (AD) [92,93,94]. Ischemic stroke is widely considered to be a significant risk factor for the incidence of vascular dementia and Alzheimer’s disease [95]. The co-occurrence of stroke and AD are prominently higher as the patient’s age increases. Circulatory miRNAs may be used as peripheral biomarkers of IS and AD. Approximately the microanalysis of 157 miRNAs isolated from the blood samples of IS patients have reported upregulation of 138 miRNAs and downregulation of 19 miRNAs; this is indicating the underlying events during IS such as hypoxia regulation, hematopoiesis, and angiogenesis. This kind of miRNA profiling is a significant approach to differentiating the pathophysiological changes between stroke-induced AD and cardio-embolic strokes [95,96]. For instance, the upregulation of miR-155 has been associated with neuroinflammation in AD and this miRNA is able to facilitate pathogenesis of AD by regulating T-cell function [97,98]. A recent research report delineated the extensive expression of miR-124 in hippocampus, which is correlated to the synaptic plasticity and memory dysfunction in mice models [99]. Overexpression of miR-98 is reported to enhance Aβ production and phosphorylation of tau in models of APP_Swe_/PS1 mice [100,101] (https://www.frontiersin.org/articles/10.3389/fphar.2019.00665/full—B39). MiR-455-3P and Mir-34a-5p expression is correlated to the Aβ1–42 levels in CSF of AD patients [102]. Furthermore, the expression of miR-125b and miR-146a are reported to be upregulated and fostered the neuronal apoptosis and tau phosphorylation in the cellular models of AD [103]. Hence, this miRNA could be considered as the potential biomarkers to decipher the stroke-related AD. Yet, there are studies to be performed to examine the role of miRNAs in the co-occurrence of stroke and AD.

### 2.8. MicroRNAs and Inflammation

Stroke-induced inflammation is associated with intricate pathological process initiating from microglial cell activation, circulating leukocytes infiltration (such as neutrophils, lymphocytes, and macrophages), and proinflammatory mediator release by ischemic and immune cells [104]. Microglial cells are the innate immune macrophages of the CNS and are activated after IS. Activated microglia and associated inflammatory factors such as TNF-α can facilitate the progression of neurodegeneration [105]. Cytokines viz., “IL-1β, IL-6, and C-reactive protein (CRP), TNF-α, nitric oxide (NO), ROS and prostanoids” are released during post-ischemic inflammation, which can further aggravate primary brain injury and IS complications [106,107]. The peripheral cytokines are generated from T-lymphocytes, mononuclear phagocytes, NK-cells and polymorphonuclear leukocytes can effectively mediate inflammation during IS [108].

During cerebral ischemia, the proinflammatory gene expressions induced for NF-κB, interferon regulatory factor-1, hypoxia-inducible factor-1 (HIF-1), and STAT3 generation. Consequently, these factors regulate the expression of cytokines and adhesion molecules, such as ICAM-1, P-selectin and E-selectin. These cellular adhesion molecules (CAM) enable leukocyte adhesion to microvascular endothelium in the cerebral ischemic area [109]. NF-κB is a heteromeric transcription factor involved in the activation of proinflammatory genes viz., *TNF-α*, *ICAM-1*, *COX-2*, *iNOS*, and *IL-6* [110,111]. Extensive rise of CAMs after ischemic stroke could foster leukocyte migration through brain endothelial cells [112]. Neutrophils targeting the ischemic tissue can produce matrix metalloproteins (MMPs) to cross the BBB. *MMP-9* and *MMP-2* are responsible for BBB breakdown and hemorrhagic transformation after ischemic stroke [113]. 

t-PA therapy against ischemic stroke triggers significant adverse effects like neurotoxicity and BBB disruption through MMPs [113]. t-PA related adverse effects are mediated through NMDA receptors, LDL receptor-related protein (LRP), activated protein-C (APC), protease-activated receptor 1, and platelet-derived growth factor C [113]. Hence, therapeutic modalities in combination therapy are promising approaches against IS to mitigate the adverse effects from tPA therapy. 

A wide array of miRNAs can target genes that are involved in post-ischemic inflammation. Studies show that miR-424 has a protective effect against focal cerebral ischemic injury by inducing impairment of microglial activation [114]. Moreover, miR-let-7c-5p confers protection against neuroinflammation following cerebral ischemia by “inhibiting microglial activation” and “translational repression of caspase-3” [115]. Upregulated expression of miR-124 can promote suppression of microglial differentiation and macrophage deactivation via C/EBP-α-PU.1 pathway. Ischemic inflammatory processes can activate toll-like receptors (TLR). Further, TLR can activate *NF-κB* which induces the expression of proinflammatory genes and adhesion molecules [116].

Thirteen (13) TLR have been identified. Among them, TLR4 signaling promotes post-ischemic inflammatory injury [117,118]. TLR4 on the surface of microglial cells is upregulated in response to hypoxia [119]. It has been shown that miR-181c downregulates TLR4 expression via 3′-UTR. Furthermore, miR-181c can suppress “NF-κB activation and associated proinflammatory products including TNF-α, IL-1β, and iNOS” [120]. Hence, the therapeutic strategies to activate miR-181c specifically may mitigate TLR and NF-kB mediated inflammatory responses during IS. MiR-155 is capable of inducing the expression of TNF-α and IL-1β in cerebral ischemia by regulating TLR4 activity, consequently suppressing the expression of inflammatory mediators viz., suppressor of cytokine signaling 1 (SOCS1) and myeloid differentiation primary response protein 88 (MyD88). These findings support the need for developing novel “miR-155-based therapeutic modality” against ischemic stroke [24]. In microglia, macrophages, and monocytes, the miR-155 expression induced in response to proinflammatory stimuli such as IFN-γ and TNF-α during ischemia-induced inflammation [121]. It was demonstrated that miR-181c can regulate posttranscriptional TNF-α production in microglia directly. Thus, miR-181c decreases the release of TNF-α from microglia cells and mitigates neuronal apoptosis during ischemic stroke [122]. MiR-181a has an anti-inflammatory effect mediated by direct regulation of IL-1α in monocytes and macrophage lines. MiR-146a can suppress the expression of *IL-1β* and *IL-6* (proinflammatory cytokines) suggesting that miR-146a has an important role in “inflammation associated with neurological diseases” including ischemic stroke; this report delineated the need to develop therapeutic modalities to enhance miR-146a expression for alleviating neuroinflammation associated with IS [27] (Figure 2). Moreover, miR-106a and miR-124 in microglia and macrophages stimulated the upregulation of IL-10 and TGF-β respectively [123,124]; Serum miR-124, miR-9, and miR-219 are reported to be decreased in acute IS, thereby facilitating neuroinflammatory responses and neuronal death during IS [31]. In addition, the specific role of various other miRNAs for modulating the pathophysiology of various risk factors (including inflammation) causing IS by affecting specific genes and mRNAs were described in Table 1, Table 2 and Table 3.

### 2.9. MicroRNAs, BBB Disruption, and Edema

Ischemic stroke with brain edema can confer a progressive enhancement in the intracranial pressure and also induce low blood flow to the brain tissue [126]. Vasogenic edema is exemplified by extravasation and BBB damage, which eventually induce extracellular fluid accumulation in cerebral parenchyma [126]. On a physiological level, the cerebral edema is mediated by the endothelial dysfunction, MMPs, and aquaporins (AQP). Several miRNAs have been implicated in stroke-induced cerebral edema via direct or indirect mechanisms [127]. Endothelial cells have multiple miRNAs involved in controlling BBB function in normal and pathological conditions [128]. Experimental models of medial cerebral artery occlusion (MCAO) have shown that the upregulation of miR-150 could increase BBB permeability; the overexpression of miR-150 in microvascular endothelial cells mitigates the expression of claudin-5, a tight junction protein; the low levels of claudin-5 eventually evokes extensive rise in “endothelial permeability” and neural cell death in oxygen-glucose deprivation condition (OGD) during stroke [129]. These effects may be reversed by downregulation of angiopoietin receptor TIE-2 (a target of miR-150); antagomir miR-150 treatment prevented BBB disruption and decreased post-ischemic degeneration, possibly via regulation of endothelial survival [129]. 

Stroke incidence can induce alteration in the expression of MMPs, which eventually disrupt endothelial integrity and increase BBB permeability [130]. For instance, *MMP-9* was expressed extensively in astrocytes, microglia, neurons, and endothelial cells, and found to be implicated in BBB disruption during IS [130]. A significant upregulation of miR-21 and *MMP-9* was observed in the rat hippocampus 24-h after cerebral ischemia; miR-21 is involved in the ERK-mediated upregulation of *MMP-9* through calcium-dependent mechanism [131]. The prospective research should focus on the development of novel therapeutic modalities to mitigate the *MMP-9* expression in ischemic stroke by modulating the expression of miR-21. 

Aquaporins are a family of proteins involved in water transport across plasma membranes and have a pivotal role in maintaining both intracellular and extracellular water homeostasis [34]. To date, 13 subtypes of AQP have been identified. AQP1, AQP4, and AQP9 are the most abundant in CNS where AQP4 exhibits the highest expression levels [132]. Spatial and temporal expression of AQP4 across the CNS depends on the stroke model. Its expression has been demonstrated to be higher during the glial-specific swelling [133]. For instance, the upregulated expression of AQP4 can be observed in perivascular end-feet across the border of ischemic lesion [134]. Substantial expression of AQP4 in astrocyte end-feet that form the BBB is considered to be a key regulator in inducing vasogenic edema following focal ischemia [135]. MiR-130a is a transcriptional repressor of AQP4 M1 and encodes an AQP4 isoform that exhibits the greatest water permeability [34]. Thus, the suppression of miR-130a-activated AQP4 M1 transcript and its protein expression results in the reduction of infarct volume during IS [34]. Interestingly, AQP4 is a target of miR-29b and the overexpression of miR-29b significantly mitigates AQP4 expression with concomitant reduction of BBB disruption, edema, and infarction volume in a mouse model of focal ischemia [33]. These findings indicate the ability of miRNAs to control spatial and temporal AQP expression in relation to post-stroke edema.

### 2.10. MicroRNAs and Neuronal Death

Neurons in the penumbral region during the IS pathogens can be recovered using some selective therapeutic interventions; but the extent of recovery depends on the elapsed time before therapy is initiated. Excitotoxicity, oxidative stress, and inflammation can foster neuronal death in the penumbra region for hours and days after IS [136]. 

It is important to note that many miRNAs target the antiapoptotic protein Bcl-2 [137]. For instance, miR-15 cluster is capable of targeting Bcl-2 during focal ischemia eventually miR-15 upregulation induces detrimental effects. The impairment of miR-15 may lead to upregulation of Bcl-2 protein expression and confer endothelial neural cell protection and subsequently mitigate infarct volume and vascular dysfunction during the focal ischemia [36]. MiR-497 also targets Bcl-2, and antagomir miR-497 treatment increases Bcl-2 levels, which is accompanied by reduction in infarction volume during ischemic stroke [138]. MiR-181a is reported to be enhanced across the vulnerable areas of focal ischemia mainly across the “hippocampal CA1 region” during global ischemia and mitigates Bcl-2 expression [137,139,140]. A therapeutic strategy to suppress miR-181a during global-cerebral ischemia significantly increases Bcl-2 levels and reduces hippocampal CA1 neuronal loss [140]. Moreover, miR-181a downregulation in primary cultured astrocytes and leads to an increase in Bcl-2 and myeloid cell leukemia 1 (Mcl-1) protein levels, and reduces mitochondrial dysfunction and apoptosis triggered by glucose deprivation [141]. 

MiR-29b suppresses multiple Bcl-2 family members, including *Bcl-2*, *Mcl-1,* and *Bcl-w* (Bcl-2) [35]. MiR-29b can impair the *Bcl-2* gene expression and enhance neuronal cell death suggesting another new therapeutic target against ischemic brain injury [35]. For instance, the expression of miR-29b significantly enhanced after “transient focal ischemia” in rat brains and promotes cortical neuronal death in cell cultures [35]. *Bcl-w* overexpression could offer protection against miR-29b-induced neuronal death, indicating that miR-29b may promote neuronal death via Bcl-w suppression and may be involved in apoptosis. A majority of miRNAs target the intrinsic apoptosis pathway: *In vitro* studies showed that miR-21, miR-25, and miR-181c can predominantly regulate TNF-α signaling through the extrinsic apoptosis pathway. Overexpression of miR-21 downregulates “Fas-ligand expression” and prevents OGD-induced neuronal death in cultured cortical neurons [122,142]. MiR-181c can suppress *TNF-α* and partially prevents neuronal death [122]. These reports suggest there is miR-181c-mediated regulation of *TNF-α* expression during ischemia/hypoxia and microglia-mediated neuronal injury for modulating the neuroinflammation [122]. The above findings have demonstrated that several miRNAs can target protein translation of both intrinsic and extrinsic apoptosis pathways involved in recovering the neurons in prenumbral region or cortical region, thereby influencing the clinical outcome of IS. 

### 2.11. MicroRNAs in Neurogenesis

Post-stroke events can orchestrate neuronal regeneration by promoting cell proliferation in the subventricular region through stem cell migration and differentiation across damaged brain regions into new neuronal phenotypes to replace the damaged cells [143]. MiR-17-92 and miR-124 are known regulators of embryonic neurogenesis. Focal ischemia can upregulate the miR-17-92 cluster expression in neural progenitor cells in adult mice; the overexpression of miR-17-92 typically enhanced proliferation in cultured progenitor cells in the subventricular zone of ischemic region [144]. 

MiR-124 is an important regulator of brain development and is expressed in mature neurons in the adult brain [145]. Focal ischemia has been shown to downregulate the expression of miR-124 in neural progenitor cells residing in the subventricular zone; miR-124 transfection reduces the ischemia-induced proliferation via suppression of Jagged-1 (JAG1) membrane protein involved in modulating Notch signaling pathway [146]. Notch signaling pathway is a crucial pathway involved in determining cell fate because this signaling plays a pivotal role in the maintenance, proliferation, and differentiation of neural stem cells across the subventricular zone after stroke [147,148]. It has been shown that miR-210 overexpression consistently promotes neurogenesis in the adult brain [149]. In addition, overexpression of miR-210 facilitates neural progenitor proliferation, which could be preferred as therapeutic modalities for clinical studies against IS [150].

### 2.12. MicroRNAs in Angiogenesis

Several miRNAs (ex. miR-210) are involved in post-ischemic angiogenesis, which is crucial for the restoration of blood flow to ischemic regions to promote functional recovery after a stroke [151]. Molecular mechanisms underlying the angiogenesis in post-traumatic conditions are associated with complex processes controlled through the angiogenic factors viz., *VEGF, FGF-2, netrins,* and *PDGF*. These factors typically can promote the endothelium synthesis, and proliferation as well as migration of pericytes consequently fosters angiogenesis through neovascularisation [152,153]. Hypoxia-induced miRNAs modulate post-stroke angiogenesis by regulating the activity of VEGF. For example, MiR-107 can enhance angiogenesis after stroke via hypoxia-induced factor 1A (*HIF-1A*) [154,155]. Furthermore, miR-107 enhances angiogenesis by inducing expression of endogenous VEGF and by suppressing Dicer-1 [156]. Furthermore, the increased expression of miR-107 could promote angiogenesis in the penumbra region; which was delineated by the administration of antagomir miR-107 that effectively reduced the capillary density in penumbra and enhanced the infarction volume after focal ischemia [156]. Upregulation of MiR-210 is also reported to promote Notch signaling to enhance angiogenesis after ischemic injury of brain [157].

A number of miRNAs are reported to be involved in angiogenesis but are not associated with VEGF. For instance, the overexpression of miR-124 initiates neurovascular changes consequently leading to angiogenesis at 8 weeks after middle cerebral artery occlusion (MCAO), typically through USp14-dependent RE1-silencing transcription factor (REST) degradation [158]. 

Therapeutic strategies that block miR-155 can enhance the “brain microvasculature” after ischemia. This was examined in animal models using miR-155 inhibitor to target miR-155 followed by the modulation of “Rheb/Akt interaction” and stabilization of “ZO-1 (zonula occludens)”, a crucial regulatory scaffolding tight junction-protein [159]. Thus, the inhibition of miR-155 after distal MCAO reduces infarction volume, supports microvascular integrity, preserves capillary tight junctions, and improves blood flow in the penumbra via targeting the GTP-binding protein RHEB that stabilizes tight junctions [159]. 

MiR-145 is reported to be involved in modulating the blood glucose metabolism and it exhibits a vital role in the diabetic MCAO rat models. An in vivo study showed that bone marrow-derived mesenchymal stem cells (BMSC) from rats with type 1 diabetes mellitus (DM-BMSC) have reported to enhance “capillary formation” and “axonal outgrowth” in cultured primary cortical neurons by mitigating the expression of miR-145. Thus, the suppression of miR-145 may be a beneficial strategy for neural tissue regeneration and functional recovery in type 1 diabetic MCAO [43].

## 3. MiRNA Profiling and Ischemic Stroke

Several miRNAs and their selective target genes are involved in the pathophysiology of IS [21]. The pathogenesis of ischemic stroke is also predominantly related to inflammation, platelet activation, atherosclerosis, and blood coagulation [21]. The miRNAs could control particular target genes, which contributes to their significant value as diagnostic markers [21]. Assessing the early signs of IS is a significant strategy to control risk factors and further to manage conditions that are related directly to stroke risk [21]. Hence, ascertaining the prominent role of miRNAs as diagnostic markers for IS pathophysiology is imperative to medical professionals to control these risk factors. 

As we have discussed about the important miRNAs, which plays a key role in arterial hypertension, a crucial risk factor for ischemic stroke [38,39,40,41]. MiR-155, miR-125a/b-5p, miR-22, and miR-487b can control blood pressure and affect patient treatment to modulate the pathogenesis of ischemic stroke. Thus, these MiRNAs involved in the development of hypertension could be identified as potential new diagnostic biomarkers for early diagnosis and assessment of ischemic stroke risk [45,46,47]. The suppression of circulating plasma miR-223 and upregulation of miR-144 has been observed in T2D patients during IS [42]. It has been shown that miR-145 levels are inversely correlated with the “neuroprotective effects” produced from BMSC treatment in diabetic rats [43]. Furthermore, hyperglycemia during diabetes can affect brain miRNA during post-stroke. In particular, studies have shown a link between miR-Let7A, *ASK-1*, and microglial function in hyperglycemia-induced oxidative stress [22]. Thus, ascertaining the miR-145 and miR-Let7A levels in diabetic patients may deliver an effective diagnosis to promote early control of diabetes-induced IS (Figure 2).

A growing body of research indicates the importance of miRNAs as regulators of inflammatory processes in the pathogenesis and progression of atherosclerosis. MiR-181a has been suggested as a therapeutic target for immune-mediated inflammation in atherosclerosis, a risk factor for ischemic stroke [62]. MiR-181a targets a set of genes associated with inflammation, and suppresses expression of *c-Fos*, which is an important transcription factor in inflammation. The role of miR-155 in the regulation of adaptive/innate immune reactions associated with stroke pathology has been explored. Dysregulation of miR-155 is associated with inflammatory evens during atherosclerosis. Expression of miR-155 can mediate the production of TNF-α, and targets CARHSP1. Moreover, “TNF-α-induced activation of NF-kB” specifically regulates miR-155 expression. Available data provide evidence that upregulated miR-155 reduces inflammation via *miR-155–CARHSP1–TNF-α* signaling [66,67]. These findings also correlate with research that demonstrates proinflammatory role of miR-155 in microglia [24,121]. Increased levels of miR-155 in microglial cells are associated with the progression of immune response and production of inflammatory mediators; this could bestow an important therapeutic approach against “inflammation” during ischemic stroke. A direct association was already reported between the expression of miR-155, and inhibition of SOCS-1, with simultaneous production of inflammatory mediators during brain injury; miR-155 dysregulation promotes inflammation in the CNS by disrupting SOCS-1 function and by increasing cytokine levels and NO production during ischemic stroke [24]. Thus, the efficacy of miR-155 may be examined using clinical studies against IS to consider as an effective diagnostic marker for inflammation-mediated IS. MiR-181c has an important role in post-stroke TNF-α-mediated neurotoxicity. Expression of miR-181c leads to a decrease in TNF-α release by microglial cells and confer neuronal death [122]. 

Furthermore, miRNA are present in the serum and plasma in large amounts and are highly stable in samples subjected to multiple freeze/thaw cycles, which makes them attractive as biomarkers. MiRNAs are abundant in the serum/plasma of both humans and animal species [160,161]. Thus, miRNA expression profiles of several experimental animal models potentially could serve as biomarkers in disease research. Past studies delineated how molecular profiles of diseases can be determined using miRNAs. Widely used miRNA profiling methods are functional bioinformatics, bioanalytics such as Next Generation Sequencing (NGS), quantitative PCR (qPCR), digital PCR (dPCR), NanoString technologies Counter© Analysis System, and mirco array-technology [21,162,163,164,165,166]. Nowadays, it is possible to use instrumentation with high sample and data throughput to record several miRNA parameters simultaneously. NGS could be considered as the preferred platform for the discovery of novel miRNAs and enables a reliable collection of both qualitative information (e.g., which miRNAs) and quantitative information (counts of miRNAs) from ischemic stroke patient samples. For routine miRNA profiling as diagnostics, it is desirable to use simpler technologies such as qPCR [167,168]. miRNA profiling can generate terabytes of transcriptomic data that can be combined with other genomic, proteomic, and metabolomic data to interpret in accordance to the available electronic health record (EHR) of stroke patients. It would be desirable if miRNAs could be used as prognostic factors in combination with other parameters such as age, onset of treatment, NIHSS score, pre-existing hypertension, heart disease, and diabetes [169]. This kind of Vertical-Omics or Multi-Omics are useful to identify the most significant genes viz., *HMGB1*, *YWHAZ*, *PIK3R1*, *STAT3*, *MAPK1*, *CBX5*, *CAPZB*, *THBS1*, *TNFRSF10B*, *RCOR1* (target genes of miRNAs) related to ischemic stroke pathogenesis underlying the inflammation, blood coagulation, atherosclerosis, etc. [21,170]. Gene ontology is another method for creating miRNA-gene targets with interacting proteins using information from available databases. For instance, miR-19a-3p is a significant miRNA that can be modulated in selected gene ontologies. This miRNA is referred to as a significant novel biomarker for ischemic stroke to be examined more fully in future studies [21,166,171,172].

A significant challenge for miRNA profiling as diagnostic makers in ischemic stroke is data interpretation [162,173,174]. There is a wide range of bioinformatics and machine learning tools that can be used to create predictive models to study long non-coding RNA (lncRNA) to miRNA to mRNA networks during IS [173,175,176,177,178]. The translation of scientific studies into everyday clinical practice is not trivial, and the broad applicability of NGS under routine conditions for miRNA profiling to understand the pathogenesis of IS has yet to be achieved. A harmonization of technical and bioinformatic approaches will continue to occupy future generations studying IS pathogenesis based on human studies. The aim is to decipher and interpret clinical results to ascertain the significance of miRNAs that are beneficial for human therapeutic applications [21,179,180,181].

MiRNA research has unlimited potential for developing future of diagnostics and personalized interventions. MiRNAs are promising clinical biomarkers for early diagnosis of acute stroke. Further research is warranted to elucidate the molecular disease mechanisms to develop safer and more effective diagnosis and therapeutic modalities for IS [182]. Myriad scientific reports have attempted to correlate variations in miRNAs expressions during stroke and their molecular targets to determine the mechanism of action. This could help to understand the pathophysiological processes associated with ischemic stroke; and how risk factors like atherosclerosis, arterial hypertension, and diabetes mellitus predispose the patients to stroke [182]. The variable patterns of miRNA expressions in stroke demonstrate the possibility of using them as potential biomarkers. There is a need to ascertain the miRNAs as diagnostic tools which may provide new ideas for developing novel effective pharmacological therapies against ischemic stroke.

## 4. Conclusions

MiRNAs have been identified as gene regulators for controlling various physiological brain functions as well as disease biomarkers of IS and stroke-mediated Alzheimer’s disease. MiRNAs can target hundreds of proteins in several regulatory *cell signaling loci* under normal and pathological conditions. Normal miRNA expression is necessary for central nervous system (CNS) development and function while miRNA dysregulation in brain cells may lead to neurovascularization, which can enhance susceptibility to stroke and other neurological dysfunctions. Several studies explained the association between “miRNA expression and post-ischemic pathophysiology such as excitotoxicity, inflammation, and neuronal death”. These reports have outlined the significant role of miRNAs in the pathophysiology of IS; these reports also delineated that certain “specific miRNAs” could be promoted for clinical trials as novel therapeutics against IS. There is evidence of miRNA (and their target genes) being involved in the angiogenesis, neurogenesis, and neuroprotection during IS. Hence, the role of miRNAs as therapeutic approaches is predicted as double-edged sword effect. In the future, miRNA profiling could become a prominent diagnostic marker for IS. The modulation of miRNAs can potentially be useful for developing new ischemic stroke diagnostic methods.

## Figures and Tables

**Figure 1 ijms-21-06728-f001:**
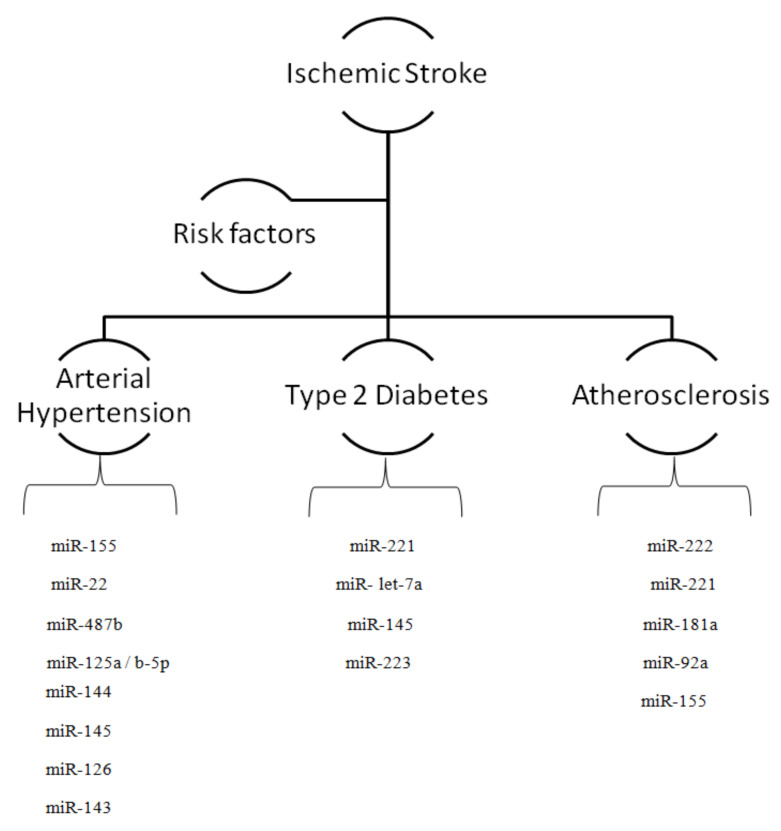
The intricate role of miRNAs in modulating risk factors associated with ischemic stroke.

**Figure 2 ijms-21-06728-f002:**
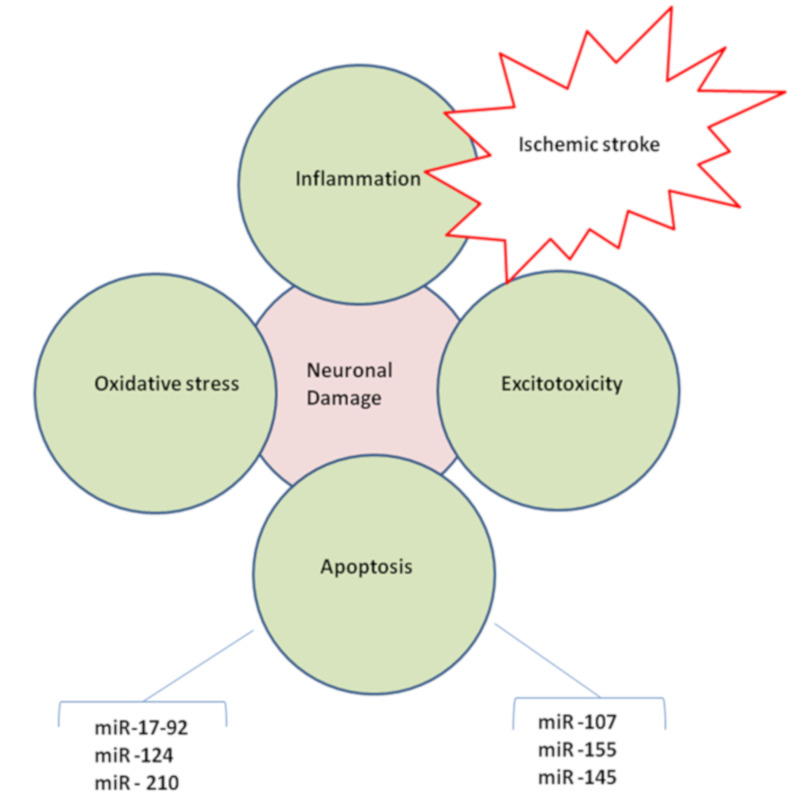
Cerebral ischemia accompanying pathogenic mechanisms viz., inflammation, oxidative stress, apoptosis, and excitotoxicity, which contribute to the neuronal damage during ischemic stroke. Potential therapeutic arenas for mitigating these pathophysiological processes could be stimulation of angiogenesis and neurogenesis through miRNA modulators.

**Table 1 ijms-21-06728-t001:** The role of miRNAs to modulate specific genes and mRNA of various risk factors of ischemic stroke.

miRNA	Target Gene	Process	Effects Produced by Modulating mRNA Role
miR-126	VCAM1 [22]	Atherosclerosis	Reduced neutrophil infiltration
miR-155	PU.1 [23]	Arterial hypertension	Reduced monocyte maturation
miR-155	SOCS1 [24]	Inflammation	Suppression of inflammation
miR-125b	NR2A [25]	Excitotoxicity	Decreased NMDA activation
miR-125b	p53 [26]	Neuronal death	Reduced neuronal death
miR-146a	IRAK-1, IL-6, IL-8 [27]	Inflammation	Reduced inflammation
miR-145	KLP4, KLP5 [28]	Atherosclerosis	Promotes SMC growth
Let-7a	Casp3 [22]	Apoptosis	Reduced apoptosis
miR-221	KIT [29]	Type 2 Diabetes	Endothelial dysfunction
miR-221	KIP1 [29]	Atherosclerosis	Promotes SMC growth
miR-222	KIP2 [29]	Atherosclerosis	Promotes SMC growth
miR-223	NR2A [30]	Excitotoxicity	Increased NMDA activation
miR-424	NRF2 [31]	Inflammation	Reduced inflammation
miR-181a miR-25	Bim (BCL2L11) [32]	Apoptosis	Reduced apoptosis
miR-29b, miR-130a	AQP4 [33,34]	Edema	Reduces edema formation
miR-29b, miR-15	Bcl-2 [35,36]	Apoptosis	Reduced apoptosis

SMC—smooth muscle cells.

**Table 2 ijms-21-06728-t002:** The specific role of miRNAs modulating several target genes involved in post-ischemic inflammation.

miRNA	Target	mRNA Activity	Effects Produced by Modulating mRNA Role
miR-424	NRF2 [31]	Increase	Suppression of inflammation
miR-let- 7c-5p	Casp3 [115]	Decrease	Suppression of inflammation
miR-124	C/EBP-α-PU.1 [125]	Decrease	Suppression of inflammation
miR-155	SOCS1, MyD88 [24]	Decrease	Suppression of inflammation
miR-106a	IL-10 [27]	-	Suppression of inflammation
miR-146a	IL-6, IL-1β [27]	Decrease	Suppression of inflammation
miR-9	MMP-9, MMP-13 [115]	Decrease	Suppression of inflammation
miR-219	MPP9 [31]	Decrease	Suppression of inflammation
miR-181c	TLR4 [120]	Increase	Suppression of inflammation
miR-181a	IL1 [122]	Increase	Suppression of inflammation

MMP—matrix metalloproteinase, IL—interleukin, TLR—Toll-like receptors, SOCS1—Suppressor of cytokine signaling 1.

**Table 3 ijms-21-06728-t003:** The significant role of miRNAs in targeting several mRNAs to modulate the production of target proteins involved in the etiology of ischemic stroke.

Risk Factors	mRNA	Target	Source
Arterial Hypertension	miR-155	PU.1	[23]
	miR-22	CHGA	[39]
	miR-487b	IGF-I	[41]
	miR-125a/b-5p	ET-1	[38]
Diabetes	miR-221	KIT	[29]
	miR-let-7a	ASK-1	[22]
	miR-145	ABCA1	[43]
	miR-223	P2Y	[42]
	miR-144	IRS-1	[42]
Atherosclerosis	miR-222	KIP2	[29]
	miR-221	KIP1	[29]
	miR-145	KLP4, KLP5	[28]
	miR-126	VCAM1	[22]
	miR-143	ABCA1	[28]
	miR-92a	Kruppel factor	[28]
	miR-155	ETS1, AT1r	[63]
	miR-181a	c-Fos	[32]

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
