# Peer review of "Can miRNAs Be Considered as Diagnostic and Therapeutic Molecules in Ischemic Stroke Pathogenesis?—Current Status"

_ijms, 2020, doi:10.3390/ijms21186728_

Round 1

Reviewer 1 Report

In this paper, the author discussed miRNAs in the Ischemic Stroke Pathogenesis. In general, the paper was well written, and the functional miRNAs were well organized in the table. Below are the minor suggestions. 1. On line 255, what types of miRNAs are involved in transient focal ischemic reperfusion and ischemic brain injury? 2. On line 305, please describe more miRNAs that are associated with Alzheimer’s disease 3. It is now well known that miRNAs are a critical component in extracellular vesicles in including exosomes. To describe the most updated knowledge, it would be better to mention the potential of exosome containing miRNAs as a novel diagnostic tool.

Author Response

On behalf of my coauthors, please accept my sincere thanks and gratitude for careful perusal and critical review of our manuscript entitled “Can miRNAs be considered as diagnostic and therapeutic molecules in Ischemic Stroke Pathogenesis? - Current Status”. We have revised the manuscript based upon the reviewers’ comments as well as your suggestion. Adequate care has been taken to accommodate each and every suggestion of the reviewers. An itemized, “point-by-point” reply to all the comments is attached separately where we have clearly presented our specific response and additions, deletions and/or modifications that have been made in the revised text, and highlighted.

Point-by-Point Answers to the Comments

 Reviewer-1

 Comment-1: In this paper, the author discussed miRNAs in the Ischemic Stroke Pathogenesis. In general, the paper was well written, and the functional miRNAs were well organized in the table.

 Response: Authors convey sincere thanks for appreciation.

 Comment-2: On line 255, what types of miRNAs are involved in transient focal ischemic reperfusion and ischemic brain injury?

Response: Authors included the text related to the miRNA expressions during TFI and ischemic brain injury in the manuscript.

Comment-3:   On line 305, please describe more miRNAs that are associated with Alzheimer’s disease

Response: Authors included the text related to the miRNA expressions associated with AD in the manuscript.

Comment-4:   It is now well known that miRNAs are a critical component in extracellular vesicles in including exosomes. To describe the most updated knowledge, it would be better to mention the potential of exosome containing miRNAs as a novel diagnostic tool.

Response: Authors included the text related to the exosomal-miRNA expressions during ischemic brain injury in the manuscript.

Reviewer 2 Report

The manuscript by Kirill V. Bulygin et al.

Title: Can miRNAs be considered as diagnostic and therapeutic approaches in Ischemic Stroke Pathogenesis? - Current Status

Authors have discussed the current state of research related with the role of miRNAs in cerebral ischemia and pathogenesis of ischemic stroke (IS). They reviewed the pros and cons of miRNAs that modulate risk factors for pathological mechanisms of IS to develop diagnostic tests and novel therapeutic options for IS.

The study is of importance for scientific community. Theoretical background is sufficient to explain the reasons why the study was conducted.

Comments for improvement of the review:

An abbreviation should be introduced at the first use and then the abbreviation should be used in the text. For example in the following sentence the term »microRNA« should be replaced with »miRNA«:    In this review, we explored the role of microRNAs…

Optional: authors may consider to modify the title, namely, microRNA is a molecule and not an approach.

Grammar should be checked and some spaces are missing, for example in the sentence:

MiR-155, miR-125a/b-5p, miR-22, and miR-487b can regulate blood pressure and thereforeaffects patient clinical outcome during IS-targeted therapy

Gene names and symbols should be edited according to the HGNC nomenclature:

https://www.genenames.org/

For example see the official gene name for the ASK1 (ASK-1) gene.

https://www.genenames.org/data/gene-symbol-report/#!/hgnc_id/6857

Gene names should be written in italics.

Table 3: for some target genes symbols are given and for the other gene names are given, this should be unified. For example, CHGA is a gene symbol for chromogranin A.

It is recommended to add one column with gene IDs to tables.

Figures could be improved, font could be unified, and more data should be added.

MicroRNA -target-disease interactions/associations could also be visualized as networks and such a figure would attract citations.

Author Response

Dear Reviewer 2:

On behalf of my coauthors, please accept my sincere thanks and gratitude for careful perusal and critical review of our manuscript entitled “Can miRNAs be considered as diagnostic and therapeutic molecules in Ischemic Stroke Pathogenesis? - Current Status”. We have revised the manuscript based upon the reviewers’ comments as well as your suggestion. Adequate care has been taken to accommodate each and every suggestion of the reviewers. An itemized, “point-by-point” reply to all the comments is attached separately where we have clearly presented our specific response and additions, deletions and/or modifications that have been made in the revised text, and highlighted.

Reviewer-2

Comment-1: The study is of importance for scientific community. Theoretical background is sufficient to explain the reasons why the study was conducted.

Response: Authors convey sincere thanks for appreciation.

Comment-2: An abbreviation should be introduced at the first use and then the abbreviation should be used in the text. For example in the following sentence the term »microRNA« should be replaced with »miRNA«:    In this review, we explored the role of microRNAs…

Response: Authors implemented changes as per the reviewer’s suggestion in the manuscript. 

Comment-3: Optional: authors may consider to modify the title, namely, microRNA is a molecule and not an approach.

Response: Authors implemented changes in the title as per the reviewer’s suggestion in the manuscript. 

Comment-4: Grammar should be checked and some spaces are missing, for example in the sentence: MiR-155, miR-125a/b-5p, miR-22, and miR-487b can regulate blood pressure and therefore affects patient clinical outcome during IS-targeted therapy

Response: Authors corrected the grammars in the sentences throughout the manuscript.

Comment-5: Gene names and symbols should be edited according to the HGNC nomenclature:

https://www.genenames.org/; For example see the official gene name for the ASK1 (ASK-1) gene. https://www.genenames.org/data/gene-symbol-report/#!/hgnc_id/6857

Response: Authors corrected the gene names and symbols with official names given in the above links.

Comment-6: Gene names should be written in italics.

Response: Authors corrected the gene names written in the italics.

Comment-7: Table 3: for some target genes, symbols are given and for the other gene names are given, this should be unified. For example, CHGA is a gene symbol for chromogranin A.

It is recommended to add one column with gene IDs to tables.

Response: Authors corrected gene symbols in the all corresponding tables. Text of the article almost crossed the word limit as per MDPI IJMS author guidelines. Hence, we minimized the text wherever possible. 

Comment-8: Figures could be improved, font could be unified, and more data should be added.

Response: Authors have improved the figures with more resolution and data was sufficiently incorporated as per the text given in the manuscript.

Comment-9: MicroRNA-target-disease interactions/associations could also be visualized as networks and such a figure would attract citations.

Response: Authors corrected gene symbols in the all corresponding tables. Text of the article almost crossed the word limit as per MDPI IJMS author guidelines. Hence, we minimized the text wherever possible. Authors apologise for that.